# MGNNI: Multiscale Graph Neural Networks with Implicit Layers

**Juncheng Liu   Bryan Hooi   Kenji Kawaguchi   Xiaokui Xiao**
National University of Singapore
{juncheng,bhooi,kenji,xiaoxk}@comp.nus.edu.sg

## Abstract

Recently, implicit graph neural networks (GNNs) have been proposed to capture long-range dependencies in underlying graphs. In this paper, we introduce and justify two weaknesses of implicit GNNs: the constrained expressiveness due to their limited effective range for capturing long-range dependencies, and their lack of ability to capture multiscale information on graphs at multiple resolutions. To show the limited effective range of previous implicit GNNs, we first provide a theoretical analysis and point out the intrinsic relationship between the effective range and the convergence of iterative equations used in these models. To mitigate the mentioned weaknesses, we propose a multiscale graph neural network with implicit layers (MGNNI) which is able to model multiscale structures on graphs and has an expanded effective range for capturing long-range dependencies. We conduct comprehensive experiments for both node classification and graph classification to show that MGNNI outperforms representative baselines and has a better ability for multiscale modeling and capturing of long-range dependencies.

## 1   Introduction

In recent years, graph neural networks (GNNs) have been widely adopted on graph-related tasks, such as node classification, link prediction, and graph classification [29]. In general, GNNs utilize both node attributes and graph topology to produce meaningful node representations for downstream applications. To achieve this, most modern GNNs follow a "message passing" mechanism: at each iteration, they iteratively aggregate representations of neighboring nodes of each node with its own representation to generate new representations. During this process, each iteration is typically parameterized as a single-layer neural network with learnable weights. Many GNN models have been proposed by adopting different aggregations techniques (e.g., GCN with renormalization [17], GAT with attentive aggregations on neighbors [27], and SGC [28] using aggregations without non-linear activation). In spite of the effectiveness achieved by the aforementioned GNNs on different tasks, they fail to effectively capture long-range information on graphs, since a GNN with $T$ layers can only capture information up to $T$ hops away.

To overcome this deficiency of previous GNNs, recent work has proposed implicit graph neural networks [13, 20, 21] to effectively capture long-range dependencies. These implicit graph neural networks generally define a fixed-point equation as an implicit layer for aggregation and generate the equilibrium $Z^*$ as the node representations. To get the equilibrium, they either use an iterative solver to solve the equation or directly obtain a closed-form solution with guaranteed convergence. Meanwhile, they utilize implicit differentiation to achieve $\mathcal{O}(1)$ memory complexity when computing the gradients during the iterations. As mentioned in Gu et al. [13], Liu et al. [20], these models can be treated as a graph neural network with infinite layers which has the same transformation and shared weights in each layer. This makes them able to effectively capture long-range dependencies without excessive memory requirements as compared with previous GNNs.

36th Conference on Neural Information Processing Systems (NeurIPS 2022).

Despite the superiority of implicit GNNs shown in several applications requiring long-range information, an important question — what the farthest range these models can capture information from — has not been studied. Although these models are usually claimed as GNNs with infinite depth [13, 20], in this paper, we first point out the *effective range* of these models (i.e., the maximum hops they can effectively capture dependencies for each node) is actually bounded by a certain value. We provide analyses on the intrinsic relationship between the effective range and the convergence of the iterative equation used in implicit GNNs. Specifically, these models usually use a contraction factor $\gamma$ to ensure the convergence of the iterative map [20, 21], which indeed exponentially decays the distant information during the aggregation at the same time. This design inherently limits the effective range of propagation and hinders their ability to capture long-range dependencies.

Besides the limited effective range, implicit GNNs also cannot effectively capture multiscale information on graphs, i.e., graph features at various scales. In contrast, several GNNs without implicit layers [30, 2, 1, 8] have been proposed to utilize multiscale information to improve the model capacity. For example, Xu et al. [30] proposes JKNet which leverages different neighborhood ranges by skip connections and adaptive aggregations of hidden representations at different layers. MixHop [1] learns neighborhood mixing relationships by mixing hidden representations at various distances. These explicit GNN models demonstrate the effectiveness of utilizing multiscale information from neighbors at various scales. However, it is still not clear how to utilize multiscale information with implicit GNNs since there are no different "layers" in implicit GNNs that can be used for capturing multiscale information at different scales.

Motivated by the above limitations of previous implicit GNNs, we propose our multiscale graph neural network with implicit layers (MGNNI) which brings multiscale modeling into implicit GNNs and expands their effective range for capturing long-range dependencies. We summarize the contributions of this work as follows:

- We introduce the concept of effective range for implicit GNNs, and provide theoretical analyses on the effective range that previous implicit GNNs can capture distant information from. We then point out that their effective range is limited although previous models are generally regarded as GNNs with infinite layers.

- We propose MGNNI as a new implicit GNN model with multiscale propagation to expand the effective range and capture underlying graph information at various scales.

- We conduct comprehensive experiments with synthetic datasets and real-world datasets on both node classification and graph classification to demonstrate that MGNNI has better performance and a better ability to capture both long-range and multiscale information compared with other baselines.

## 2 Related work

**Implicit Models**  Implicit neural networks use implicit hidden layers which are *implicitly* defined: the outputs are determined by the solutions of some underlying equations. A notable advantage of these implicit models is that they can generally backpropagate through the fixed-point solution using implicit differentiation to achieve *constant* memory complexity regardless of the "depth" of the network. There is an emerging interest in implicit layers in recent years [4, 3, 5, 11]. To name a few, Bai et al. [4] propose the deep equilibrium model (DEQ) demonstrating the ability of implicit models in sequence modeling; Multiscale DEQ (MDEQ) [5] brings multiscale modeling into implicit deep networks for image classification and semantic segmentation. Kawaguchi [15] analyses the global convergence of deep linear implicit models and Geng et al. [11] provide a gradient estimate for implicit models to avoid the costly exact gradient computation.

**Graph Neural Networks**  GNNs have been widely used in different tasks for graph-structured data. Even with different aggregation schemes (e.g., skip connection [30, 8] and attention [27]), convolutional GNNs [17, 28, 30] generally involve finite aggregation layers (usually less than 20 layers) with different learnable weights, which makes them unable to effectively capture long-range dependencies. Although RevGNN [19] is proposed with 1000 layers, it has to use deep reversible architectures [12], which requires excessive amount of time for training. Inspired by implicit models [4, 5] on image and text data, implicit graph neural networks [13, 20, 21] have been proposed to capture long-range information with constant memory complexity. Implicit GNNs generally define an

aggregation equation and obtain the fixed-point solution of the equation as the outputs. In particular, Gu et al. [13] propose IGNN where they ensure the well-poseness and use iterative solvers to obtain fixed-point solutions. Liu et al. [20] propose EIGNN as a linear implicit GNNs where a closed-form solution is derived. Park et al. [21] construct an input-dependent linear iterative map for predicting the properties of a graph system. However, these implicit GNNs cannot model multiscale information in underlying graphs. In contrast, several explicit GNNs, such as JKNet [30], MixHop [1], and N-GCN [2], have shown that multiscale information is helpful to improve the model capability. To fill the gap, our model MGNNI brings multiscale modeling to implicit GNNs.

## 3 Preliminaries

A graph is represented as $\mathcal{G} = (\mathcal{V}, \mathcal{E})$ which contains the node set $\mathcal{V}$ with $n$ nodes and the edge set $\mathcal{E}$. In practice, graph neural networks take the adjacency matrix $A \in \mathbb{R}^{n \times n}$ and the node feature matrix $X \in \mathbb{R}^{m \times n}$ of $\mathcal{G}$ as input data. For simplicity, considering unweighted adjacency matrix $A$, then $A_{i,j} = 1$ if $(i,j) \in \mathcal{E}$, for any two nodes $i, j \in \mathcal{V}$; otherwise $A_{i,j} = 0$. Given input graph data $(\mathcal{G}, X)$, depending on different classification tasks, graph neural networks are required to provide a prediction $\hat{y}$ for a node or a graph to match the true label $y$.

**Aggregations in GNNs**   GNNs typically employ a trainable aggregation process that iteratively pass the information from each node to its adjacent nodes, followed by a non-linear activation. Without loss of generality, a typical aggregation step at layer $l$ can be written as follows:

$$Z^{(l+1)} = \phi(W^{(l)} Z^{(l)} S + \Omega^{(l)} X), \tag{1}$$

where $Z^{(l)} \in \mathbb{R}^{h_l \times n}$ is the hidden states in the layer $l$ which stacks the state vectors of every nodes denoted as $z^{(l)} \in \mathbb{R}^{h_l}$; $S \in \mathbb{R}^{n \times n}$ is the normalized adjacency matrix; $W^{(l)} \in \mathbb{R}^{h_{l+1} \times h_l}$ and $\Omega^{(l)} \in \mathbb{R}^{h_{l+1} \times m}$ are the matrices of trainable weight parameters; $\phi$ denotes a non-linear activation function. Recently proposed GNN models use different forms of this graph aggregation process. For example, simplified graph convolution [28] removes the non-linear activation, use only one weight matrix $W$, and sets $Z^{(0)} = X$ and $\Omega = 0$.

In addition to GNNs with explicitly defined layers, GNNs with implicit layers [13, 20, 21] also follow a similar aggregation form, but with tied weight matrices $W$ and $\Omega$ at each iteration step. For these implicit GNNs, the aggregation step is generally changed to $Z^{(l+1)} = \phi(W Z^{(l)} S + \Omega X)$. Given such an aggregation step, implicit GNNs can be seen as iterating the aggregation step an infinite number of times until convergences. To ensure the convergence, IGNN [13] enforces $\|W\|_\infty \leq \kappa/\lambda_{pf}(A)$ with $\kappa \in [0, 1)$, where $\lambda_{pf}$ is the Perron-Frobeius (PF) eigenvalue [7]. EIGNN [20] and CGS [21] instead achieve this by introducing a contraction factor $\gamma$ in the aggregation step.

Using EIGNN as an example, it defines the aggregation as an iterative mapping with a contraction factor $\gamma$ as follows:

$$Z^{(l+1)} = \gamma g(F) Z^{(l)} S + X, \tag{2}$$

where $\gamma \in [0, 1)$ and $g(W)$ is a bounded mapping that projects the trainable weight matrix $F$ into a Frobenius norm ball of radius $< 1$. Given the iterative mapping, implicit GNNs obtain the equilibrium states $Z^* = \phi(W Z^* S + \Omega X)$ as the final representations by using either root-finding approaches or closed-form solutions.

## 4 Effective range of previous implicit GNNs

Implicit deep learning is considered as a method to increase the effective depth of deep neural networks [4, 24]. Previous works on implicit GNNs (including IGNN [13] and EIGNN [20]) claim that these models can be viewed as an infinite-layer GNN. However, in this section, we point out that the effective range within which previous implicit GNNs can capture the long-range information is actually bounded. In other words, their abilities for capturing long-range dependencies are still restricted. We provide analyses revealing the intrinsic relationship between the effective range and the convergence of the iterative equation in implicit GNNs. In addition, we also provide empirical results to support our analyses.

To investigate the effective range, we first provide an analysis on sensitivity, i.e., how changes in node features of node $p$ affect the equilibrium of a distant node $q$.

**Theorem 1.** *Given two nodes $p$ and $q$ that are $h$-hops apart, using Equation (2) for propagation, if we perturb node features $X_{:,p}$ of node $p$ by $\Delta X_{:,p} \in \mathbb{R}^m$, the L2 norm of the change in node $q$'s equilibrium $\|\Delta Z^*_{:,q}\|$ is upper bounded as follows:*

$$\|\Delta Z^*_{:,q}\| \leq \frac{\gamma^h}{1 - \gamma}\|g^h(F)\Delta X_{:,p}S^h_{p,q}\|. \tag{3}$$

The complete proof can be found in Appendix A.1.

At first glance, the norm of the change in equilibrium $\|\Delta Z^*_{:,q}\|$ would not be zero no matter how large the distance between node $p$ and $q$. However, Theorem 1 shows that $\|\Delta Z^*_{:,q}\|$ decays exponentially with distance along the graph. Therefore, in practice, this change will fairly quickly fall below the roundoff error in floating-point numbers or the stopping criterion in the iterative solver used to obtain the fixed-point solution, then the change $\Delta X_{:,p}$ on node $p$ cannot affect the equilibrium of node $q$. This is the intuition about what we call the effective range. To formalize it, we provide the definition of $\theta$-effective range as follows:

**Definition 1.** For any given error parameter $\theta > 0$, the $\theta$-effective range $h$ is the maximum integer such that exists some pairs of nodes $p$ and $q$ that are $h$-hop apart, when node features $X_{:,p}$ of node $p$ are perturbed by $\Delta X_{:,p}$, the L2 norm of the change in node $q$'s equilibrium $\|\Delta Z^*_{:,q}\| > \theta$.

By Definition 1, we know that, given any $h' > h$, for **all pairs** of node $p$ and $q$ that are $h'$-hop apart, the L2 norm of the change $\|\Delta Z^*_{:,q}\| \leq \theta$, which means the equilibrium of node $q$ cannot be affected in practice. To analyse the $\theta$-effective range, we can derive the corollary of Theorem 1:

**Corollary 1.** *With Equation (2) for propagation, given any error constant $\theta > 0$, the $\theta$-effective range $h$ is upper bounded: $h < \frac{\ln(\theta(1-\gamma))}{\ln \gamma}$. Therefore, if node features $X_{:,p}$ of node $p$ are perturbed, the perturbation can only affect the equilibrium of nodes which are up to $\frac{\ln(\theta(1-\gamma))}{\ln \gamma}$-hop away from $p$.*

The complete proof can be found in Appendix A.2. Note that the above analysis is directly applicable to two recent Implicit GNN models with a contraction factor $\gamma$, i.e., EIGNN [20] and CGS [21].

Asides from the above analysis, we further verify the theoretical analysis with synthetic experiments on the same chain dataset as in EIGNN [20] and IGNN [13], where the task requires simply passing information from one end to the other of a chain graph. See Appendix C.1 for detailed settings. We use the basic form of EIGNN [20] with iterative methods (i.e., iterating Equation (2) to find the equilibrium) as the model. We follow the same experimental setup in EIGNN and then perturb node features by masking the features of the starting node $p$ to all zeros. To support our theoretical analysis, we investigate how the change of node $q$'s equilibrium (i.e., $\Delta Z^*_{:,q}$) behaves as $q$ gets farther away from $p$. In Figure 1, we show that $\|\Delta Z^*_{:,q}\|$ decays as node $q$ gets further from $p$ in terms of the distance. For example, with $\gamma = 0.5$, $\|\Delta Z^*_{:,q}\|$ becomes numerically 0 when $q$ and $p$ are around 25 hops apart, which indicates that node $q$ can no longer receive any information from node $p$ at this distance.

The above theoretical analysis and empirical evidences show how different values of $\gamma$ can constrain the expressiveness of implicit GNNs, i.e., the range within which they can effective capture long-range dependencies. In short, smaller $\gamma$ results in a shorter effective range of implicit GNNs. A straightforward strategy to expand the effective range is to increase $\gamma$ (e.g., to close to 1). However, a large $\gamma$ can cause instability and difficulty for the convergence of iterative mapping, which would empirically compromise the efficiency of iterative solvers as found in our experiments. We provides the empirical evidences for this in Appendix B. This raises the question: *how can we capture longer range dependencies while ensuring the convergence of the iterative mapping?* Our multiscale MGNNI approach aims to answer this question.

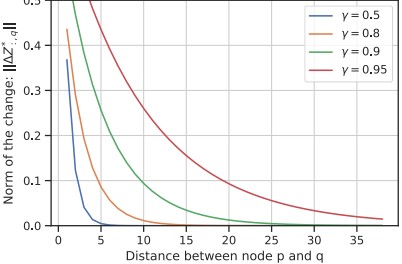

Figure 1: The norm of the change of equilibrium $\|\Delta Z^*_{:,q}\|$ decays as node $q$ becomes further from $p$.

# 5 Multiscale implicit graph neural networks

Previous implicit GNNs [13, 20, 21] propagate the information from 1-hop neighbors while applying a decay with the contraction factor $\gamma$ in each iterative step (e.g., using Equation (2)). As analysed in Section 4, this design inherently limits the effective range of propagation since the information decays exponentially as the range grows linearly. Besides the limited effective range, previous implicit GNNs also cannot capture multiscale information on graphs similarly to explicit GNNs, such as JKNet [30] and MixHop [1], which combine information at different scales of the graph.

Motivated by these limitations, we propose multiscale graph neural networks with implicit layers (MGNNI) which can first expand the effective range to capture long-range dependencies and then capture information from neighbors at various distances. MGNNI contains multiple propagation components with different scales and learns a trainable aggregation mechanism for mixing latent information at various scales.

## 5.1 The MGNNI model

A single $m$-scale propagation module in MGNNI model is defined as the following iterative mapping:

$$Z^{(l+1)} = \gamma g(F) Z^{(l)} S^m + f(X, \mathcal{G}), \tag{4}$$

where $\gamma \in [0, 1)$ and $m$ denotes a hyperparameter for the graph scale (i.e., the power of adjacency matrix). $f(X, \mathcal{G})$ is a parameterized transformation on input features and graphs, and $g(F)$ is normalized weight matrix defined as:

$$g(F) = \frac{1}{\|F^\top F\|_{\mathrm{F}} + \epsilon_F} F^\top F \tag{5}$$

with an arbitrary small $\epsilon_F > 0$. Note that in multiscale propagation, at each iterative step, the model can capture the information along with a m-step path, while previous implicit GNNs only consider 1-hop neighbors. In this way, MGNNI is able to capture dependencies within a longer range over iterations. Now we provide the analysis to show that the iterative mapping in MGNNI (i.e., Equation (4)) converges to a unique equilibrium $Z^*$:

$$\lim_{l \to \infty} Z^{(h)} = Z^* \quad \text{s.t.} \quad Z^* = \gamma g(F) Z^* S^m + f(X, \mathcal{G}). \tag{6}$$

**Theorem 2.** *Given the bounded damping factor $\gamma \in [0, 1)$, the proposed iterative map for propagation (i.e., Equation (4)) is a contraction mapping and the unique fixed-point solution $Z^*$ can be obtained by iterating Equation (4).*

This can be proved by using the properties of matrix vectorization and the Kronecker product with the Banach fixed Point Theorem. The complete proof is given in Appendiex A.3.

**Multiscale propagation** With a set of multiple scales $M = \{m_1, ..., m_k | m_i \neq m_j \forall i, j\}$, we can have multiple propagation modules and obtain $k$ equilibriums with different scales $\{Z^{*1}, Z^{*2}, ..., Z^{*k}\}$. Given those equilibriums, we propose a scale-aggregation mechanism utilizing learnable attentions, as to learn the contributions of different scales for each node automatically through the learning objective. For each node $i$, $z_i^{*t}$ denotes $t$-th equilibrium in $Z^{*t}$ and the attention value $\beta_i^t$ is defined as follows:

$$\beta_i^t = q^T \tanh(W_a z_i^{*t} + b_a), \tag{7}$$

where $q$ is the parameterized attention weight vector, $W_a$ and $b_a$ are the weight matrix and the bias vector, respectively. Given attention values for different scales $\{\beta^1, ... \beta^k\}$, the final weights are normalized by softmax function:

$$\alpha_i^t = \text{softmax}(\beta_i^t) = \frac{\exp(\beta_i^t)}{\sum_{j=1}^k \exp(\beta_i^j)}. \tag{8}$$

Larger $\alpha_i^t$ indicates that the corresponding scale is more important for node $i$. The final embeddings $Z'$ are obtained by aggregating the equilibriums at different scales with corresponding weights:

$$z_i' = \sum_{t=1}^k \alpha_i^t z_i^{*t}, \tag{9}$$

$$\hat{Y} = f_o(Z'), \tag{10}$$

where the predictions $\hat{Y}$ are generated by a problem-specific decoding function $f_o$.

## 5.2 Expanded range via multiscale propagation

In multiscale propagation, nodes receive information from further $m$-hop neighbors rather than only immediate neighbors, which enlarges the effective range of message passing. It is similar with a larger receptive field in convolutional neural networks. We prove that the effective range for receiving distant information is enlarged by using multiscale propagation.

**Theorem 3.** *Given two nodes $p$ and $q$ are $h$-hop apart, using propagation with $m$-hop neighbors (i.e., Equation ([4](#)), if we perturb node features $X_{:,p}$ of node $p$ by $\Delta X_{:,p} \in \mathbb{R}^m$, the L2 norm of the change in node $q$'s equilibrium $\|\Delta Z_{:,q}^*\|$ is upper bounded as follows:*

$$\|\Delta Z_{:,q}^*\| \le \frac{\gamma^{\frac{h}{m}}}{1-\gamma}\|g^{\frac{h}{m}}(F)\Delta X_{:,p}S_{p,q}^h\|. \tag{11}$$

The complete proof is provided in Appendix A.5.

Similar to Corollary 1, we analyse the effective range of multiscale propagation by considering when the change in the equilibrium of node $q$ becomes smaller than a certain numerical error.

**Corollary 2.** *Using propagation with $m$-hop neighbors (i.e., Equation ([4](#)), given any small error constant $\theta$, the $\theta$-effective range $h < \frac{m \ln(\theta(1-\gamma))}{\ln \gamma}$. Hence, the perturbation on node features of node $p$ can affect the equilibrium of nodes which are located up to $\frac{m \ln(\theta(1-\gamma))}{\ln \gamma}$-hop away from $p$.*

This is the corollary of Theorem 3. The proof is given in Appendix A.5. Under the same condition, the effective range is expanded by using multiscale propagation to consider $m$-hop neighbors in a propagation step.

## 5.3 Training MGNNI

To train MGNNI, we can simply iterate Equation ([4](#)) until it converges to the equilibrium $Z^*$ for the forward pass. We do not use closed-form solutions as in Liu et al. [20] since it would slow the training with large graphs and a large number of node features. For backward pass, given a loss $\ell$, we can use implicit differentiation to compute the gradients of trainable parameters by directly differentiating through the equilibrium $Z^*$ by:

$$\frac{\partial \ell}{\partial (\cdot)} = \frac{\partial \ell}{\partial Z^*} \left(I - J_\varphi(Z^*)\right)^{-1} \frac{\partial \varphi(Z^*, X, \mathcal{G})}{\partial (\cdot)}, \tag{12}$$

where $Z^* = \varphi(Z^*, X, \mathcal{G}) = \gamma g(F)Z^*S^m + f(X, \mathcal{G})$ and $J_\varphi(Z^*) = \frac{\partial \varphi(Z^*, X, \mathcal{G})}{\partial Z^*}$. We provide the complete derivation in Appendix A.6. One of advantages of directly differentiating through $Z^*$ is that the memory consumption is only one layer regardless the number of iterative steps in forward pass. In contrast, differentiating over iterative steps requires large memory to store intermediate variables.

Note that $(I - J_\varphi(Z^*))^{-1}$ in Equation ([12](#)) is expensive to compute due to the computation of the Jacobian $J_\varphi(Z^*)$ and the inverse. However, we can solve a linear equation with a Vector-Jacobian product (VJP) to achieve cheaper computation of $\frac{\partial \ell}{\partial Z^*}\left(I - J_\varphi(Z^*)\right)^{-1}$:

$$u^T = u^T J_\varphi(Z^*) + \frac{\partial \ell}{\partial Z^*}. \tag{13}$$

Note that, the VJP $u^T J_\varphi(Z^*)$ can be efficiently computed by automatic differentiation packages (e.g., PyTorch [22]) without forming the Jacobian. Subsequently, the gradients $\frac{\partial \ell}{\partial (\cdot)}$ can be obtained.

## 5.4 Discussion and comparison with previous implicit GNNs

In general, compared to previous implicit GNNs (i.e., IGNN [13], EIGNN [20], and CGS [21]), our model MGNNI brings multiscale modeling to implicit GNNs, which allows the model to capture graph information of different granularities. The multiscale idea has shown its effectiveness on explicit GNN models, such as N-GCN [2], JKNet [30], and MixHop [1]. However, to our knowledge, there is no previous implicit GNNs able to capture graph information at various scales. Specifically, MGNNI essentially has multiple implicit layers with different focuses on various scales coexisting side by side.

**Computational complexity** MGNNI has similar time complexity with IGNN and CGS as they all use iterative methods to iterate Equation (4) until convergence. The asymptotic time complexity is $O(K(h^2n + hn^2))$ where $h$ is the number of hidden units after the input transformation $f(X, \mathcal{G})$ and $K$ is the number of iterations in an iterative method. In contrast, EIGNN costs $O(n^3)$ to conduct eigendecomposition for the adjacency $S$, which is costly and prohibitive for large graphs. Additionally, for training, it requires $O(h_i^3 + h_i^2 n)$ to get the closed-form solution and $O(h_i^3)$ to conduct eigendecomposition for the weights, where $h_i$ is the number of input features. Comparing MGNNI and EIGNN, MGNNI is more efficient on large graphs as it utilizes iterative solvers for fixed-point solutions, whereas EIGNN requires eigendecomposition to get closed-form solutions.

The above analysis of computational complexity mainly considers the process of propagation. Here, we discuss the time complexity of some additional operations in these implicit GNNs, e.g., the attention mechanism in MGNNI and the projection on the weight matrix in IGNN, which generally cost much less time compared with the main propagation process. The attention mechanism used in MGNNI has the time complexity $O(h'hn + hn)$ (according to Equation (7)), where $h'$ is the number of hidden units in the attention module. Similarly, IGNN also has some additional operations, e.g., it requires a projection of the weight matrix $W$ in each training iteration to ensure the well-posedness condition $\|W\|_\infty \leq \kappa/\lambda_{\mathrm{pf}}(A)$. It needs $O(n^2)$ to get and modify the maximum row sum of $W$ (i.e., $\|W\|_\infty$). Overall, MGNNI has the same level of time complexity compared with IGNN and CGS.

# 6 Experiments

In this section, we show that MGNNI can effectively capture long-range dependencies and mixed graph information at various scales. Therefore, MGNNI provides better performance on both node classification and graph classification compared with representative baselines. Specifically, we conduct the experiments[1] on 7 datasets for node classification (including 1 synthetic and 6 real-world datasets: Color-counting, Cornell, Texas, Wisconsin, Chameleon, Squirrel, PPI) and 6 datasets (MUTAG, PTC, PROTEINS, NCI1, IMDB-Binary, IMDB-Multi) for graph classification. As we follow the same experimental settings on some datasets, we reuse the results of some baselines from literatures. Descriptions of datasets and experimental settings are detailed in Appendix C.

## 6.1 Experiments with synthetic datasets

**Color-counting dataset** We construct the synthetic dataset for node classification. The graph contains several chains and some nodes on each chain have different colors. The color information is encoded in node features. The nodes on the same chain share the same label which is the majority color appeared on this chain. The model is supposed to predict the majority color on a chain, which requires the model able to capture the long-range dependencies and also count the occurrence of each color. In Figure 2, we compare MGNNI with different scales against previous implicit GNN models (i.e., IGNN, EIGNN, and CGS). MGNNI (M={1,4,8}) denotes that three scales with m=1, m=4, and m=8 are used. MGNNI with higher exponents (i.e., M={1,4,8}) generally performs better than IGNN and CGS, which indicates that

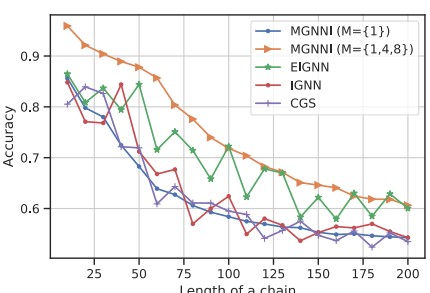

Figure 2: Averaged accuracies on color-counting dataset.

multiscale propagation with higher-hop neighbors can effectively expand the effective range for long-range dependencies. Note that MGNNI (M={1}) can be regarded as MGNNI with only single scale. It is outperformed by MGNNI (M={1,4,8}), which again shows the effectiveness of multiscale propagation. MGNNI (M={1}), IGNN and CGS are all outperformed by EIGNN, suggesting that implicit GNNs with iterative solvers may still suffer from approximation error issues. We also provide the comparison between MGNNI and representative explicit GNNs in Appendix C.1 (See Figure 5). It further demonstrates the superiority of MGNNI over explicit GNNs in terms of the ability of capturing long-range dependencies.

---

[1]The implementation can be found at https://github.com/liu-jc/MGNNI

Table 1: Results on heterophilic graph datasets: mean accuracy (%) ± stdev over different data splits.

| | Cornell | Texas | Wisconsin | Chameleon | Squirrel |
|---|---|---|---|---|---|
| **# Nodes** | 183 | 183 | 251 | 2,277 | 5,201 |
| **# Edges** | 280 | 295 | 466 | 31,421 | 198,493 |
| **# Classes** | 5 | 5 | 5 | 5 | 5 |
| Geom-GCN [23] | 60.81 | 67.57 | 64.12 | 60.90 | 38.14 |
| SGC [28] | 58.91 ± 3.15 | 58.92 ± 4.32 | 59.41 ± 6.39 | 40.63 ± 2.35 | 28.4 ± 1.43 |
| GCN [17] | 59.19 ± 3.51 | 64.05 ± 5.28 | 61.17 ± 4.71 | 42.34 ± 2.77 | 29.0 ± 1.10 |
| GAT [27] | 59.46 ± 6.94 | 61.62 ± 5.77 | 60.78 ± 8.27 | 46.03 ± 2.51 | 30.51 ± 1.28 |
| APPNP [18] | 63.78 ± 5.43 | 64.32 ± 7.03 | 61.57 ± 3.31 | 43.85 ± 2.43 | 30.67 ± 1.06 |
| JKNet [30] | 58.18 ± 3.87 | 63.78 ± 6.30 | 60.98 ± 2.97 | 44.45 ± 3.17 | 30.83 ± 1.65 |
| GCNII [8] | 76.75 ± 5.95 | 73.51 ± 9.95 | 78.82 ± 5.74 | 48.59 ± 1.88 | 32.20 ± 1.06 |
| H2GCN [34] | 82.22 ± 5.67 | 84.76 ± 5.57 | 85.88 ± 4.58 | 60.30 ± 2.31 | 40.75 ± 1.44 |
| IGNN [13] | 61.35 ± 4.84 | 58.37 ± 5.82 | 53.53 ± 6.49 | 41.38 ± 2.53 | 24.99 ± 2.11 |
| EIGNN [20] | 85.13 ± 5.57 | 84.60 ± 5.41 | **86.86 ± 5.54** | 62.92 ± 1.59 | 46.37 ± 1.39 |
| CGS [21] | 68.11 ± 9.41 | 62.97 ± 9.23 | 63.53 ± 9.81 | 40.57 ± 1.61 | 31.78 ± 0.89 |
| MGNNI | **85.95 ± 6.10** | **84.86 ± 5.91** | 86.67 ± 4.31 | **63.93 ± 2.21** | **54.50 ± 2.10** |

Besides the experiments on Color-counting dataset, following Liu et al. [20], we also conduct synthetic experiments on Chains dataset. MGNNI and EIGNN both maintain 100% test accuracy, demonstrating the advantages of capturing long-range dependencies. The results and detailed analyses can be found in Appendix C.1 and Figure 4.

## 6.2 Experiments with real-world datasets

**Node classification** Apart from the synthetic experiments, following Pei et al. [23], we also conduct experiments on 5 heterophilic datasets (Cornell, Texas, Wisconsin, Chameleon, and Squirrel) for node classification. On heterophilic graphs, the nodes with different class labels tend to be connected, which requires models to aggregation information from distant nodes. These graph datasets are web-page graphs of the corresponding universities or the corresponding Wikipedia pages. We use the standard train/test/val splits as in Pei et al. [23]. See the detailed setting in Appendix C.2.

The results are shown in Table 1. MGNNI generally achieves the best performance on most datasets. Comparing MGNNI and EIGNN, MGNNI provides better results, especially on Chameleon and Squirrel, which indicates that MGNNI has the superior ability of capturing long-range dependencies and multiscale information. Among implicit GNN baselines, EIGNN outperforms IGNN and CGS, which might be attributed to the less approximation error in the closed-form solution of EIGNN. Among explicit GNN baselines, H2GCN and GCNII are usually better than others, suggesting that the aggregation design in H2GCN and deeper models with residual connections are helpful on heterophilic graphs.

In addition, we also evaluate MGNNI on Protein-Protein Interaction (PPI) dataset which have multiple graphs. On each graph, proteins are presented as nodes and edges are formed if there is an interaction between two proteins. The task is to predict node labels on multi-label multi-graph inductive setting. The same train/val/test split are used as in Hamilton et al. [14]. Table 2 reports the micro-F1 scores of MGNNI against other baseline models. Compared to IGNN and EIGNN, MGNNI achieves 1.1% and 0.7% absolute improvement respectively by effectively capturing underlying multiscale information and long-range dependencies between proteins. Unlike IGNN having 4 implicit layers sequentially stacked, on PPI dataset, MGNNI resorts to parallel equilibrium layers with different scales, which makes MGNNI more efficient than IGNN. We report the efficiency comparison between MGNNI and other implicit GNNs in Appendix C.4.

Table 2: Multi-label node classification on PPI: Micro-F1 (%).

| Method | Micro-F1 |
|---|---|
| GCN [17] | 59.2 |
| GraphSAGE [14] | 78.6 |
| SSE [9] | 83.6 |
| GAT [27] | 97.3 |
| JKNet [30] | 97.6 |
| IGNN [13] | 97.6 |
| EIGNN [20] | 98.0 |
| MGNNI | **98.7** |

Table 3: Mean accuracy (%) ± stdev over 10 folds on real-world datasets for graph classification.

| | MUTAG | PTC | PROTEINS | NCI1 | IMDB-B | IMDB-M |
|---|---|---|---|---|---|---|
| **# Graphs** | 188 | 344 | 1113 | 4110 | 1000 | 1500 |
| **Avg # Nodes** | 17.9 | 25.5 | 39.1 | 29.8 | 19.8 | 13.0 |
| **# Classes** | 2 | 2 | 2 | 2 | 2 | 3 |
| GCN [17] | 85.6 ± 5.8 | 64.2 ± 4.3 | 76.0 ± 3.2 | 80.2 ± 2.0 | - | - |
| GIN [31] | 89.0 ± 6.0 | 63.7 ± 8.2 | 75.9 ± 3.8 | **82.7 ± 1.6** | 75.1 ± 5.1 | 52.3 ± 2.8 |
| DGCNN [33] | 85.8 | 58.6 | 75.5 | 74.4 | 70.0 | 47.8 |
| FDGNN [10] | 88.5 ± 3.8 | 63.4 ± 5.4 | 76.8 ± 2.9 | 77.8 ± 1.6 | 72.4 ± 3.6 | 50.0 ± 1.3 |
| IGNN [13] | 89.3 ± 6.7 | 70.1 ± 5.6 | 77.7 ± 3.4 | 80.5 ± 1.9 | - | - |
| EIGNN [20] | 88.9 ± 1.1 | 69.8 ± 5.3 | 75.9 ± 6.4 | 77.5 ± 2.2 | 72.3 ± 4.3 | 52.1 ± 2.9 |
| CGS [21] | 89.4 ± 5.6 | 64.7 ± 6.4 | 76.3 ± 6.3 | 77.2 ± 2.0 | 73.1 ± 3.3 | 51.1 ± 2.2 |
| MGNNI | **91.9 ± 5.5** | **72.1 ± 2.8** | **79.2 ± 2.9** | 78.9 ± 2.1 | **75.8 ± 3.4** | **53.5 ± 2.8** |

**Graph classification**    Besides node classification, we conduct experiments on graph classification tasks, using four bioinformatics datasets (MUTAG, PTC, PROTEINS, NCI) [32] and two social-network datasets (IMDB-Binary and IMDB-Multi). 10-fold Cross-validation is conducted as [31] and the averaged accuracies with standard deviations are reported in Table 3. The results of baselines are borrowed from Gu et al. [13] and Park et al. [21]. In general, compared with other implicit and explicit baseline models, MGNNI achieves the best performance on 3 out of 4 bioinformatics datasets and both two social-network datasets. This shows that the ability to capture long-range dependencies and multiscale information is helpful for graph classification and can be generalized to test graphs in the inductive setting.

## 6.3   Ablation Study

To further investigate the effectiveness of multiscale modeling and the contributions of different scales, we conduct the ablation study using different MGNNI variants with various scale combinations. As shown in Table 4, compared to variants with a single scale (i.e., {1}, {2} and {3}), multiscale variants generally achieve better performance on all three datasets. This verifies that multiscale modeling is effective and plays an important role in capturing graph information at various scales. Comparing variants with a single scale, the variant utilizing 1-hop propagation performs best on PPI and Chameleon, whereas the variant

Table 4: Performance of different scales.

| Scales | PPI | Chameleon | Texas |
|---|---|---|---|
| {1} | 98.35 | 61.46 | 81.35 |
| {2} | 94.62 | 58.24 | 82.97 |
| {3} | 88.56 | 56.07 | 83.78 |
| {1,2} | 98.67 | 63.93 | 83.24 |
| {1,2,3} | 98.74 | 63.75 | 84.86 |

with only 3-hop propagation obtains the best performance on Texas. This demonstrates that information at a certain scale can be more important than others on graphs with different properties. Merely considering 1-hop propagation as in previous implicit GNNs might lead to sub-optimal performance.

Besides the overall performance of different variants, in Figure 3, we also demonstrate the attention values of 100 randomly sampled nodes using MGNNI with scales $\{1, 2, 3\}$ on PPI and Chameleon. Most of the nodes on chameleon tend to have high attention weights on the first scale and rarely have attentions on the third scale, whereas some nodes on PPI have relatively higher attention values on the third scale than on the first scale. Moreover, compared to Chameleon, there are more nodes on PPI prefer the second and the third scale. These phenomena verify the effectiveness of the attention module in MGNNI, and show that different nodes might prefer information at different scales

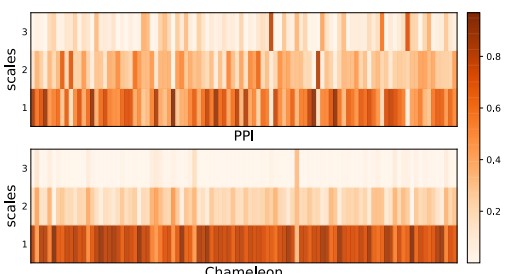

Figure 3: Attention weights of nodes at different scales on PPI and Chameleon.

according to the graph property. Additionally, we also conduct the ablation study on the effect of the attention mechanism in MGNNI (see Appendix C.5).

# 7 Conclusion

In this paper, we provide theoretical analyses on the constrained expressiveness of previous implicit GNNs due to the limited effective range for capturing distant information. We propose MGNNI which has an expanded effective range and the ability to learn mixing graph information at various scales. With synthetic experiments, we show that MGNNI with a large effective range has a better capacity to capture long-range dependencies. On various real-world datasets for both node classification and graph classification, MGNNI demonstrates superior performances compared with representative baselines, showing the effectiveness of multiscale modeling. Furthermore, the ablation study also shows that MGNNI allows different nodes to have different scale preferences, which plays an important role in adaptively capturing graph information at various scales.

Although MGNNI has the advantages of capturing multiscale graph information, MGNNI also has the limitation on potential approximation errors introduced in the iterative method as IGNN and CGS. EIGNN gets rid of these errors by using a closed-form solution with eigendecomposition which prohibits its usage on large graphs. How to mitigate approximation errors while ensuring the scalability on large graphs could be an interesting topic for future work.

## Acknowledgments and Disclosure of Funding

We would like to gratefully thank the insightful feedback and suggestions from the anonymous reviewers. We also appreciate their engagement during the discussion period. This work was supported by Proximate Beta (Grant No. A-8000177-00-00). The views and conclusions contained in this paper are those of the authors and should not be interpreted as representing any funding agencies.

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
