# Appendices

## A  Proofs

### A.1  Proof of Theorem 1

*Proof.* We denote the perturbed node features as $X'$ and change Equation (2) to the following equivalent form:

$$Z^{(k)} = \gamma^k g^k(F) Z^{(0)} S^k + \sum_{i=0}^{k-1} \gamma^i g^i(F) X' S^i.$$

Let $k \to \infty$, the first term becomes zero as $\gamma < 1$. We further decompose the above equation to $Z^{(k)} = \sum_{i=0}^{h-1} \gamma^i g^i(F) X S^i + \sum_{i=h}^{k-1} \gamma^i g^i(F) X S^i$. Since node $p$ and $q$ are h-hop apart, $S_{p,q}^i = 0$ when $i < h$. Then, we have $Z^{(k)} = \sum_{i=h}^{k-1} \gamma^i g^i(F) X' S^i$.

Let the perturbed features $X'_{:,p} = X_{:,p} + \Delta X_{:,p}$, we have $\left(X' S^i\right)_{:,q} = \left(X S^i\right)_{:,q} + \Delta X_{:,p} S_{p,q}^i$. Then we have the following:

$$\Delta Z_{:,q}^{(k)} = \sum_{i=h}^{k-1} \gamma^i g^i(F) \left(\Delta X_{:,p} S_{p,q}^i\right) \tag{14}$$

Apply the L2 norm on the change $\Delta Z_{:,q}^{(k)}$,

$$\|\Delta Z_{:,q}^{(k)}\| = \sum_{i=h}^{k-1} \gamma^i \|g^i(F) \Delta X_{:,p} S_{p,q}^i\| \tag{15}$$

$$\leq \left(\sum_{i=h}^{k-1} \gamma^i\right) \|g^h(F) \Delta X_{:,p} S_{p,q}^h\| \tag{16}$$

$$\leq \frac{\gamma^h - \gamma^k}{1 - \gamma} \|g^h(F) \Delta X_{:,p} S_{p,q}^h\| \tag{17}$$

The last inequality is derived by the sum of a geometric series.

As $k \to \infty$, $Z^* = \lim_{k \to \infty} Z^{(k)}$ and $\lim_{k \to \infty} \gamma^k = 0$. Then we have the following upper bound:

$$\|\Delta Z_{:,q}^*\| \leq \frac{\gamma^h}{1 - \gamma} \|g^h(F) \Delta X_{:,p} S_{p,q}^h\| \tag{18}$$

$\square$

### A.2  Proof of Corollary 1

*Proof.* Given the $\theta$-effective range $h$, combining Definition 1 and Equation (3) in Theorem 1, we have $\theta < \Delta Z_{:,q}^* \leq \frac{\gamma^h}{1-\gamma} \|g^h(F) \Delta X_{:,p} S_{p,q}^h\|$. Since $(1 - \gamma) > 0$, this is equivalent to $\gamma^h > (1 - \gamma)\theta$ and thus $h \ln \gamma > \ln((1 - \gamma)\theta)$. Since $\ln \gamma < 0$, we have that

$$h < \frac{\ln(\theta(1 - \gamma))}{\ln \gamma}. \tag{19}$$

Therefore, if node features $X_{:,p}$ of node $p$ are perturbed, the perturbation can only affect the equilibrium of nodes which are up to $\frac{\ln(\theta(1-\gamma))}{\ln \gamma}$-hop away from $p$.

$\square$

### A.3  Proof of Theorem 2

*Proof.* For any matrix $M \in \mathbb{R}^{m_1 \times m_2}$, we define the vectorization of the matrix by $\text{vec}[M] \in \mathbb{R}^{m_1 m_2}$, and the Frobenius norm of the matrix by $\|M\|_F$. We define the map $\varphi$ by $\varphi(Z) = \gamma g(F) Z S^m +$

$f(X, \mathcal{G})$. Recall that $Z \in \mathbb{R}^{h \times n}$. We want to show that the map $\varphi$ is contraction. Using the property of the vectorization and the Kronecker product,

$$\text{vec}[\varphi(Z)] = \gamma \, \text{vec}[g(F)ZS^m] + \text{vec}[f(X, \mathcal{G})] = \gamma[(S^m)^\top \otimes g(F)] \, \text{vec}[Z] + \text{vec}[f(X, \mathcal{G})].$$

Therefore, for any $Z, Z' \in \mathbb{R}^{h \times n}$,

$$
\begin{aligned}
\|\varphi(Z) - \varphi(Z')\|_F &= \|\text{vec}[\varphi(Z)] - \text{vec}[\varphi(Z')]\|_2 \\
&= \|\gamma[(S^m)^\top \otimes g(F)](\text{vec}[Z] - \text{vec}[Z'])\|_2 \\
&\leq \gamma \|[(S^m)^\top \otimes g(F)]\|_2 \|\text{vec}[Z] - \text{vec}[Z']\|_2 \\
&= \gamma \|[(S^m)^\top\|_2 \|g(F)]\|_2 \|\text{vec}[Z] - \text{vec}[Z']\|_2 \\
&\leq \gamma \|Z - Z'\|_F.
\end{aligned}
$$

Since $\gamma \in [0, 1)$, this shows that $\varphi$ is a contraction mapping on the metric space $(\mathbb{R}^{h \times n}, \hat{d})$ where $\hat{d}(Z, Z') = \|Z - Z'\|_F$. Thus, using the Banach fixed-point theorem, the desired statement of this theorem follows. $\qquad\square$

## A.4 Proof of Theorem 3

We denote the perturbed node features as $X'$ and change Equation (4) to the following equivalent form:

$$Z^{(k)} = \gamma^k g^k(F) Z^{(0)} S^{mk} + \sum_{i=0}^{k-1} \gamma^i g^i(F) X' S^{im}.$$

Following the similar procedure in the proof of Theorem 1, as $k \to \infty$ and $S_{p,q}^i = 0$ when $i < h$, we have $Z^{(k)} = \sum_{i=\lfloor h/m \rfloor}^{k-1} \gamma^i g^i(F) X' S^{im}$.

Let the perturbed features $X'_{:,p} = X_{:,p} + \Delta X_{:,p}$, we have $\left(X' S^i\right)_{:,q} = \left(X S^i\right)_{:,q} + \Delta X_{:,p} S_{p,q}^i$. Then we have the following:

$$\Delta Z_{:,q}^{(k)} = \sum_{i=\lfloor \frac{h}{m} \rfloor}^{k-1} \gamma^i g^i(F) \left(\Delta X_{:,p} S_{p,q}^{im}\right) \tag{20}$$

$$\tag{21}$$

Apply the L2 norm on the change $\Delta Z_{:,q}^{(k)}$,

$$\|\Delta Z_{:,q}^{(k)}\| = \sum_{i=\lfloor \frac{h}{m} \rfloor}^{k-1} \gamma^i \|g^i(F) \Delta X_{:,p} S_{p,q}^{im}\| \tag{22}$$

$$\leq \frac{\gamma^{\frac{h}{m}} - \gamma^k}{1 - \gamma} \|g^{\frac{h}{m}}(F) \Delta X_{:,p} S_{p,q}^h\| \tag{23}$$

As $k \to \infty$, $Z^* = \lim_{k \to \infty} Z^{(k)}$ and $\lim_{k \to \infty} \gamma^k = 0$. Then we have the following upper bound:

$$\|\Delta Z_{:,q}^*\| \leq \frac{\gamma^{\frac{h}{m}}}{1 - \gamma} \|g^{\frac{h}{m}}(F) \Delta X_{:,p} S_{p,q}^h\| \tag{24}$$

## A.5 Proof of Corollary 2

*Proof.* Similarly to the proof of Corollary 1 above, if $h$ satisfies that $\frac{\gamma^{h/m}}{1-\gamma} > \theta$, then the numerical error $\theta$ does not dominate. Since $(1 - \gamma) > 0$ and $\ln \gamma < 0$, this is equivalent to

$$h < \frac{m \ln(\theta(1 - \gamma))}{\ln \gamma}. \tag{25}$$

Therefore, the change on node $p$'s features can affect the equilibrium of node $q$ which is up to $\frac{m \ln(\theta(1-\gamma))}{\ln \gamma}$-hop away from $p$. $\qquad\square$

### A.6 Derivation of the gradients

Here, we provide the derivation of Equation (12) for obtaining gradients of trainable parameters with implicit differentiation Using the chain rule, the gradients of trainable parameters can be computed by:

$$\frac{\partial \ell}{\partial (\cdot)} = \frac{\partial \ell}{\partial Z^*} \frac{\partial Z^*}{\partial (\cdot)}, \tag{26}$$

where $(\cdot)$ denotes any trainable parameters within or before the implicit layer $Z^* = \varphi(Z^*, X, \mathcal{G})$. Note that $\frac{\partial \ell}{\partial (\cdot)}$ is directly handled by automatic differentiation (autodiff) packages, while $\frac{\partial Z^*}{\partial (\cdot)}$ cannot be directly obtained with autodiff since $Z^*$ and $(\cdot)$ are implicitly related.

By differentiating the both side of the fix-point equation, we can have:

$$\frac{\partial Z^*(\cdot)}{\partial (\cdot)} = \frac{\partial \varphi(Z^*, X, \mathcal{G})}{\partial Z^*} \frac{\partial Z^*(\cdot)}{\partial (\cdot)} + \frac{\partial \varphi(Z^*, X, \mathcal{G})}{\partial (\cdot)}, \tag{27}$$

where we use $Z^*(\cdot)$ to denote the case where we treat $Z^*$ as an implicit function of variables we are differentiating with respect to (e.g., the parameters of $\varphi$) and $Z^*$ alone to refer to the value of equilibrium.

By rearranging the above equation, we can obtain the explicit expression for $\frac{\partial Z^*}{\partial (\cdot)}$:

$$\frac{\partial Z^*(\cdot)}{\partial (\cdot)} = (I - J_\varphi(Z^*))^{-1} \frac{\partial \varphi(Z^*, X, \mathcal{G})}{\partial (\cdot)}, \tag{28}$$

where $J_\varphi(Z^*) = \frac{\partial \varphi(Z^*, X, \mathcal{G})}{\partial Z^*}$.

Combining Equation (28) and (26), we can have the following:

$$\frac{\partial \ell}{\partial (\cdot)} = \frac{\partial \ell}{\partial Z^*} (I - J_\varphi(Z^*))^{-1} \frac{\partial \varphi(Z^*, X, \mathcal{G})}{\partial (\cdot)}. \tag{29}$$

## B Inefficiency of using higher $\gamma$

Using a large contraction factor $\gamma$ usually make the process of finding the fixed-point more instable and difficult and it requires more iterations to find the fixed-point solution, which compromises the efficiency. We conduct the empirical experiments to verify this. We use EIGNN [20] with iterative solvers as the model on the chain dataset (as described in Sec 4). Table 5 demonstrates the training time of using different values of $\gamma$. We can see that when using $\gamma = 0.9$ can cost 2x training time compared with using $\gamma = 0.8$. Smaller $\gamma$ empirically causes faster convergence to get the fixed-point solutions.

| $\gamma$ | 0.6 | 0.8 | 0.9 | 0.95 |
|---|---|---|---|---|
| Time per epoch | 0.86s | 1.65s | 3.71s | 3.95s |
| Total time | 4596s | 8267s | 18580s | 19916s |

Table 5: Training time with different $\gamma$ used in the iterative method.

Besides EIGNN, IGNN also faces instability in the training if it uses a large contraction factor. Different with Equation (2), IGNN projects the weight matrix $W$ with a contraction factor $\kappa$ onto a convex constraint set to ensure the convergence of iterative mapping. In their official implementation repo [2], they mention that too large $\kappa$ may cause the non-convergence of the equilibrium equation, which leads significant performance degradation. In the training log they provided [3], with $\kappa = 0.98$, we can see that the loss suddenly jump to more than 2000 from around 0.019 and the accuracy degrades from 0.96 to 0.39. This verifies again that a large contraction factor may cause instability during the training, although EIGNN and IGNN use different ways for contraction.

---

[2]https://github.com/SwiftieH/IGNN/issues/3
[3]https://github.com/SwiftieH/IGNN/files/7052441/PPI_output_log.txt

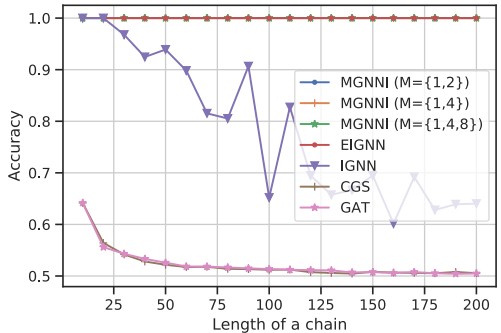 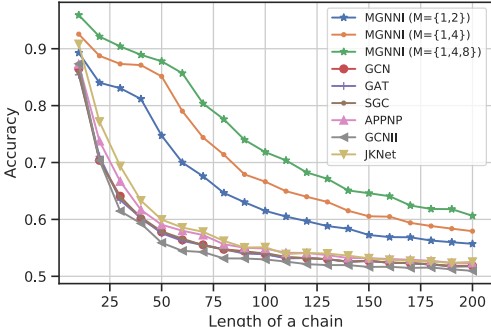

Figure 4: Averaged accuracies on Chains dataset: compare MGNNI to implicit and explicit GNNs.

Figure 5: Averaged accuracies on color-counting dataset: compare MGNNI with explicit GNNs.

## C More on Experiments

### C.1 Synthetic experiments

**Chains dataset** The synthetic chain dataset is used in Gu et al. [13] and Liu et al. [20] to evaluate the ability of models to capture distant information. Chain dataset contains several chains directed from one end to the other with length $l$. The label information is only encoded as the node features in the starting end node of each chain. The nodes on the same chains share the same class label. The task is to classify nodes into different classes, which requires simply passing information from one end to the other end of a chain graph. We consider binary classification, 20 chains for each class, and $l$ nodes in each chain. Then, the chain dataset has $40 \times l$ nodes. We randomly split the dataset for train/val/test as 5%/10%/85%.

In Figure 1 of Section 4, the starting end node is regarded as node $p$. We perturb the node features of starting node $p$ by masking the features to all zeros. After that, we measure the L2 norm of the change in node $q$'s equilibrium as increasing the distance between node $q$ and $p$ to support our analysis.

Here, as in Liu et al. [20] and Gu et al. [13], we also provide the averaged results over 20 runs on the Chains dataset for comparison between MGNNI and other representative baselines in Figure 4. As we can see, MGNNI with different scale combinations can all achieve 100% accuracy as EIGNN, which indicates the superiority of capturing long-range dependencies. In contrast, IGNN has decreasing accuracies as the chain length increases. CGS has similar performance with GAT, which demonstrates the deficiency in capturing long-range information. We conjecture the reason is that CGS uses a single layer attention variant of graph network (GN) [6] as the input transformation which is a finite GNN as GAT. For simplicity, we omit more results of other explicit GNNs as those results can provided by Liu et al. [20] in their Figure 1. Note that those explicit GNNs all generally perform worse than IGNN and EIGNN.

**More results on color-counting dataset** Besides the comparison between MGNNI and other implicit GNNs, we also compare MGNNI with explicit GNN models. Figure 5 shows that MGNNI with different scales consistently outperform all explicit GNNs, which confirms that MGNNI as an implicit GNN model has the better ability to capture long-range dependencies compared with explicit GNNs. Explicit GNN models still can achieve more than 85% accuracy when the chain length is 10. It is because that the test set are randomly sampled where some test nodes may be placed nearby the starting end node. However, the performance of explicit GNNs drops quickly as the chain length increases.

### C.2 Node classification on real-world datasets

**Dataset descriptions** We first use 5 heterophilic graph datasets as in Pei et al. [23] to evaluate the capability of capturing long-range dependencies:

- **Chameleon and Squirrel**: these graphs are originally collected by Rozemberczki et al. [25], using the web pages in WikiPedia of the corresponding topic. Nodes represent web pages and edges are

hyper-links from a web page to another. The class labels are generated by Pei et al. [23]. There are 5 categories indicating the amount of the average monthly traffic of web pages.

- **Cornell, Texas, and Wisconsin**: these datasets contain the web-page graphs of the corresponding universities. Label classes indicate the category of web pages, where 5 classes are considered, i.e., student, faculty, course, project, and staff. These three datasets are collected by the CMU WebKB project [4]. The preprocessed version generated by Pei et al. [23] is used in our experiments.

To evaluate the model capacity on multi-label multi-graph inductive setting, we conduct the experiment on Protein-Protein Interaction (PPI) dataset. The dataset is originally collected from the Molecular Signatures Database [26] by Hamilton et al. [14]. PPI dataset has 24 graphs, where each graph represents a different human tissue. Each graph has nodes representing proteins and edges indicating interactions between proteins. Each node can have maximum 121 labels which represents gene ontology sets. We use the same data splits as in Hamilton et al. [14], i.e., 20 graphs for training, two graphs for testing, and two other graphs used for validation.

**Experimental settings**   For heterophilic graphs, we compare MGNNI with 3 implicit GNNs (i.e., IGNN [13], EIGNN [20], and CGS [21]) and 8 explicit GNNs (i.e., Geom-GCN [23], SGC [28], GCN [17], GAT [27], APPNP [18], JKNet [30], GCNII [8], and H2GCN [34]). As we follow the exact same setting as in [20], we reuse their results of baselines, except CGS. For CGS and MGNNI, we conduct the experiments with 20 different runs and report the averaged accuracies with standard deviation.

For network architectures used in MGNNI on heterophilic graphs, we use two-layer MLP followed by the ReLU function as input features transformation $f(\cdot)$ and a linear map as the output transformation function (i.e., $f_o(X) = WX$). The hyperparameter search space is set as follows: multiscale set $M$ {{1,2}, {1,3}, {1,2,3}}, weight decay {5e-6, 5e-4}, learning rate {0.01, 0.05, 0.1, 0.5}. We use $\gamma = 0.8$ for all scale modules and 0.5 as the dropout rate. The Adam optimizer [16] is used for optimization. For CGS, we use the suggested network architectures (i.e., a single layer attention variant of graph network (GN) [6] and the same number of hidden neurons as in CGS paper [21]. For other hyperparameter tuning, we optimize over learning rate {0.001, 0.005, 0.01, 0.05}, weight decay {5e-6, 5e-4}, and $\gamma$ {0.5, 0.8}.

For PPI datasets, we use a 4-layer MLP directly after the multiscale propagation, while IGNN applies 4 MLPs between four consecutive IGNN layers. We set {1,2} as the multiple scales in our propagation module, and use 0.001 as the learning rate. No dropout is used.

### C.3   Graph classification on real-world datasets

**Dataset descriptions**   We conduct experiments on 4 bioinformatics datasets (MUTAG, PTC, PROTEINS, NCI1) and 2 social-network datasets (IMDB-Binary and IMDB-Multi), following identical settings as in [31, 13]. MUTAG is a dataset having 188 graphs representing mutagenic aromatic and heteroaromatic nitro compounds with 7 discrete labels. PTC is a dataset of 344 chemical compounds reporting the carcinogenicity for male and female rats and it has 19 discrete labels. PROTEINS is a dataset where nodes are secondary structure elements (SSEs) and an edge between two nodes indicates they are neighbors in the amino-acid sequence or in 3D space. It has 3 discrete labels, representing sheet, helix or turn. NCI1 is a subset of balanced datasets of chemical compounds screened for ability to suppress or inhibit the growth of a panel of human tumor cell lines with 37 discrete labels and it is made publicly available by the National Cancer Institute (NCI).

IMDB-Binary and IMDB-Multi are social-network datasets, indicating movie collaborations. Each graph contains a ego-graph for each actor/actress, where nodes represent actors/actresses and edges connect two actors/actresses if they appear in the same movie. Labels are pre-specified genres of movies and each graph has the label corresponding to its genre. The task requires models to classify the genre.

**Experimental settings**   We compare MGNNI with several representative baselines, including several explicit GNNs: Graph Convolution Network (GCN) [17], Deep Graph Convolutional Neural Network (DGCNN) [33], Fast and Deep Graph Neural Network (FDGNN) [10], and Graph Isomor-

---

[4]http://www.cs.cmu.edu/ webkb/

phism Network (GIN) [31], and two other implicit GNNs: Implicit Graph Neural Network (IGNN) [13] and Convergent Graph Solver (CGS) [21].

Since we follow the same experimental settings as in [13, 31], we reuse the results of baselines from [13, 21], except EIGNN [20]. For the network architectures used in MGNNI for graph classification, we use three-layer MLP followed the ReLU function as input feature transformation $f(\cdot)$. After the multiscale propagation with attention mechanism, we use the sum-pooling aggregator to obtain the graph representations. A linear map is used as the output transformation function (i.e., $f_o(X_g) = W X_g$). We set the number of hidden state as 32, $y = 0.8$ for all scale modules. The search space for the other hyperparameters is set as follows: multiscale set $M$ {{1,2}, {1,3}}, weight decay {0, 5e-6}, and learning rate {0.001, 0.01}. The Adam optimizer [16] is used for optimization. For EIGNN, after their implicit layer, we use a 3-layer MLP with ReLU activation followed the sum-pooling layer to obtain the graph representations. A 2-layer MLP is used for generating the final predictions. $\gamma$ is set to 0.8 and other hyperparameters (learning rate and weight decay) are tuned over the same search space.

## C.4 Efficiency Comparison

Here, we provide the efficiency comparison among IGNN, EIGNN, and MGNNI on PPI dataset. Table 6 demonstrates the training time per epoch and the total pre-processing time of different models. We can see that IGNN requires around 30s for training an epoch, while EIGNN and MGNNI requires only around 2s for an epoch. The reason of inefficiency in IGNN is that IGNN requires 4 implicit layers sequentially stacked, which means that every iterative solver needs to wait the fixed-point solution provided by the previous iterative solver for solving its own solution. In contrast, MGNNI have parallel equilibrium layers with different scales, where each equilibrium layer can get the fixed-point solution simultaneously. Thus, the training time per epoch of MGNNI is similar with that of EIGNN which only has one implicit layer. Additionally, MGNNI does not require pre-processing time to conduct eigendecomposition of the adjacency $S$ as EIGNN which may be costly for large graphs.

Table 6: Training time per epoch on PPI.

| Method | Train Time | Pre-process |
|--------|-----------|-------------|
| IGNN   | 32.7s     | N.A.        |
| EIGNN  | 2.3s      | 45s         |
| MGNNI  | 2.6s      | N.A.        |

## C.5 The effect of the attention mechanism

To quantitatively investigate the effect of the attention mechanism in MGNNI, we conduct additional experiments by removing the attention mechanism and instead use average pooling for fusing information from multiple scales. The experimental results are provided in Table 7. We can see that, if we replace the attention mechanism with average pooling, the performance would drop. It verifies the effectiveness of our attention mechanism, which is also demonstrated in Figure 3 and its corresponding explanations in Section 6.3.

Table 7: Performance of different scales.

| Scales | PPI | Chameleon | Texas |
|--------|-----|-----------|-------|
| (wo/ att) {1,2} | 98.35 | 61.46 | 81.35 |
| (w/ att) {1,2}  | 94.62 | 58.24 | 82.97 |
| (wo/ att) {1,2} | 98.67 | 63.93 | 83.24 |
| (w/ att) {1,2}  | 98.74 | 63.75 | 84.86 |