# OpenReview forum: "MGNNI: Multiscale Graph Neural Networks with Implicit Layers"
_NeurIPS.cc/2022/Conference — NeurIPS 2022 Accept_

### Official Review · Reviewer_Ctvi · 2022-07-10

**Rating:** 7
**Confidence:** 5
**Soundness:** 4 excellent
**Presentation:** 4 excellent
**Contribution:** 3 good

**Summary:**

This paper proposes a new variant of implicit graph neural network MGNNI, which is a new type of graph neural networks that captures the long-range dependencies in underlying graphs. The authors first conduct theoretical analysis on the two weaknesses of previous implicit graph neural networks, namely limited effective range for capturing distant information and constrained ability of capturing multiscale information. They then show that MGNNI with multiscale propagation could expand the effective range and capture various-scale graph information. Experiments on both synthetic dataset and real-world graphs demonstrate the effectiveness of MGNNI.

**Questions:**

- Could IGNN/EIGNN/CGS with jumping knowledge connection effectively capture multiscale information on graphs? This is a interesting baseline to further demonstrate the effectiveness of the design of MGNNI.
- How do implicit graph neural networks such as IGNN that don’t rely on contraction factor \gamma to ensure convergence be included in the theoretical analysis framework?
- How do the multiscale propagation and attention mechanism used in MGNNI affects its computational analysis?
- Does MGNNI face the over-smoothing problem?
- How does MGNNI perform on massive graphs and other types of graphs such as social networks?


**Limitations:**

Limitations are addressed in the Conclusion, but the assumption that used in theoretical analysis should be also highlighted.

No potential negative societal impact is discussed, which is fair for a methodological contribution to a well-studied problem.

**Strengths And Weaknesses:**

Strength:
- The paper is easy to follow with a clear motivation and closely related to the NeurIPS community with comprehensive experimental and theoretical analysis.
- Theoretical analysis comparison on the effective range of previous implicit GNNs and MGNNI are important and sufficient.
- The current paper might have certain impact as it further explores a mild trend in the literature, which investigates the theoretical analysis of implicit graph neural networks.

Weakness:
- The theoretical analysis has the assumption that a contraction factor \gamma is used to ensure convergence in the aggregation step, which is not used in all implicit graph neural networks. This should be highlight in limitations.
- The computational complexity for IGNN and CGS should be included for clearer comparison. How the multiscale propagation and attention mechanism used in MGNNI affects its computational analysis is missed.
- While the implicit graph neural networks are also known for effectively alleviate the over-smoothing problem for many GNNs, how MGNNI handles this over-smoothing problem should be discussed.
- For experiments, I noticed that the benchmarks used for evaluation are relatively small graphs with nodes less than 10K. While the efficiency of MGNNI is claimed as an advantage over many other SOTA implicit graph neural networks, I am curious about how the performance is on larger datasets such as OGB.
- Only four small-scale bioinformatics datasets are used in the graph classification evaluation and it is a known problem with those small and noisy datasets. It is encouraged to include other types of graphs such as social networks for more convincing.

---

> ### Author Response · Authors · 2022-08-02
> **Response to Reviewer Ctvi (Part 1)**
>
> [Response split into two posts: 1/2] We thank reviewer Ctvi for the valuable feedback and suggestions. We post our response below.
>
> **Q1.** "Could IGNN/EIGNN/CGS with jumping knowledge connection effectively capture multiscale information on graphs? This is an interesting baseline to further demonstrate the effectiveness of the design of MGNNI."
>
> They cannot directly use jumping knowledge connection. Jumping knowledge connection (JKNet) basically fuses hidden representations of all *explicit* layers. For these implicit GNNs, they only have one *implicit* layer rather than several explicit layers. Thus, we cannot directly use jumping knowledge connection for capturing multiscale information. A part of our contributions is that our work brings multiscale modeling into implicit GNNs, which has not been investigated in previous works.
>
> **Q2.** "How do implicit graph neural networks such as IGNN that don’t rely on contraction factor \gamma to ensure convergence be included in the theoretical analysis framework?"
>
> We agree that our theoretical analysis framework can include CGS and EIGNN, but cannot directly include IGNN that doesn't have the contraction factor $\gamma$. However, IGNN has a similar parameter $\kappa$ and IGNN ensures $\left\lVert W\right\rVert_{\infty} \leq \kappa / \lambda_{\mathrm{pf}}(A)$ at each training iteration. This operation can also be viewed as a contraction operation controlled by $\kappa$. When $\kappa$ is smaller, the influence between two nodes will decay faster.
> To empirically verify this property for IGNN, we conduct an additional experiment with the setting which is the same as the experiment used to get Figure 1 in our submission. The following results demonstrate the norm of the change of equilibrium $\left\lVert\Delta Z_{:, q}^{*}\right\rVert$ for IGNN also decay as the distance between node $q$ and node $p$ gets larger.
>
> | Distance | 10  | 20 | 30 |
> | ----------------- | ----- | --------- | ------ |
> | $\kappa=0.9$   | $0.3879$ | $0.1146$  | $0.0339$ |
> | $\kappa=0.8$  | $0.2083$ | $0.0036$ | $5.62\times 10^{-5}$ |
> | $\kappa=0.5$ | $1.93\times 10^{-7}$ | $0.00$ | $0.00$ |
>
> We totally agree that a more general theoretical analysis framework which can include IGNN is worth exploring. We leave it as future work and will state this in our limitations.
>
> **Q3.** "How do the multiscale propagation and attention mechanism used in MGNNI affects its computational analysis?"
>
> The attention mechanism used in MGNNI has the time complexity $O(h'hn+hn)$ (according to Equation (7)), where $h'$ is the number of hidden units in attention module. However, we would like to point out that IGNN also has some additional operations, e.g., it requires a projection of W in each training iteration to ensure the well-posedness condition $\left\lVert W\right\rVert_{\infty} \leq \kappa / \lambda_{\mathrm{pf}}(A)$ . It needs roughly $O(n^2)$ to get and modify the maximum row sum of $W$ (i.e., $\left\lVert W \right\rVert_{\infty}$). And this operation may not be easily optimized by GPU (see IGNN official implementation: https://github.com/SwiftieH/IGNN/blob/main/nodeclassification/utils.py#L203).
> Multiscale Propagation would increase the complexity of $O(K(h^2n+hn^2))$ to $O(k'K(h^2n+hn^2))$ if we use $k'$ different scales. However, remind that $K$ is the number of iterations in an iterative method (usually $K$ is around 100 in IGNN, CGS, and MGNNI). Thus, we believe this increase by $k'$ would not increase the whole time complexity much considering we usually use $k' \leq 3$ in our experiments.
>
> In conclusion, we think that only considering the main matrix multiplication part $O\left(K\left(h^{2} n+h n^{2}\right)\right)$ is still fair for the comparison among these models. We will add the above discussion to the final version of our paper.
>
> **Q4.** "Does MGNNI face the over-smoothing problem?"
>
> According to the better performance on synthetic datasets and real-world datasets, we believe MGNNI does not face the over-smoothing problem as EIGNN. We notice that in Figure 5 of appendix C.5 of EIGNN (https://proceedings.neurips.cc/paper/2021/file/9bd5ee6fe55aaeb673025dbcb8f939c1-Supplemental.pdf), they show that explicit GNNs with setting the number of layers as the chain length still perform badly on chain dataset, whereas EIGNN performs well. Then, they claim that EIGNN does not face over-smoothing problem. As MGNNI also perform as well as EIGNN on chain dataset (see Figure 4 in Appendix C of our submission), we think MGNNI does not face over-smoothing problem (at least empirically).

---

> > ### Author Response · Authors · 2022-08-02
> > **Response to Reviewer Ctvi (Part 2)**
> >
> > [Response split into two posts: 1/2] (continute on Q4)
> >
> > In our humble opinion, theoretically analysing the extent of over-smoothness for implicit GNNs family is not trivial. In fact, there are only few work [1] trying to theoretically analyse over-smoothing problem even for explicit GNNs. Theoretical analysis regarding over-smoothness for implicit GNNs is out of the scope of our paper. But we agree that it would be an interesting direction and we leave it as future work.
> >
> > [1] Deli Chen, Yankai Lin, Wei Li, Peng Li, Jie Zhou, Xu Sun. Measuring and Relieving the Over-smoothing Problem for Graph Neural Networks from the Topological View. AAAI 20.
> >
> > **Q5.** "How does MGNNI perform on massive graphs and other types of graphs such as social networks?"
> >
> > We mainly use the same datasets (including PPI dataset) and follow the experimental setting as in IGNN and EIGNN. Note that PPI dataset has multi graphs and around 57k nodes in total, which is not too small. On PPI, MGNNI outperforms other implicit GNNs as shown in Table 2 of our submission.
> >
> > As suggested, we conduct the experiments on two additional social network datasets (IMDB-B and IMDB-M) for graph classification. The results are as follows:
> >
> > | Model | IMDB-B   | IMDB-M |
> > | ----------------- | ----- | --------- |
> > | CGS   | $73.1 \pm 3.3$  | $51.1 \pm 2.2$ |
> > | MGNNI  | $75.8 \pm 3.4$ | $53.1 \pm 2.8$ |
> >
> > The results of CGS are obtained from CGS paper and we use the results of their best variants. We can see that MGNNI still outperforms CGS on these two social network datasets.
> >
> > Besides, we also conduct the preliminary experiments on ogbn-arxiv as suggested. We compare MGNNI with IGNN and EIGNN. We found that EIGNN cannot finish the training on ogbn-arxiv as it requires eigendecomposition on the whole adjacency matrix, which is time-consuming. Comparing IGNN and MGNNI, MGNNI achieves 71.09% accuracy, while IGNN obtains 70.23%. MGNNI generally costs 30.3 seconds for training an epoch, while IGNN needs around 27.4 seconds. We can see, in this case, MGNNI has better accuracy and a slightly longer training time compared with IGNN. However, before arriving at any conclusion regarding this, we need to have more comprehensive experiments for implicit GNNs on massive graphs. Due to limited time, we leave it for future work.
> >
> > We thank you for your valuable feedback again. Hope our response clarifies your questions.

---

> ### Author Response · Authors · 2022-08-07
> **Look forward to your reply**
>
> Dear Reviewer Ctvi,
>
> First, we appreciate your valuable feedback and suggestions again. We would like to kindly remind you that we have posted our response with more explanations and experimental results to your questions.
>
> Hope you are satisfied with our response. And we are looking forward to your reply. If you still have any concerns, we are also willing to discuss/clarify further.

---

> > ### Comment · Reviewer_Ctvi · 2022-08-08
> > **Thank you for your response**
> >
> > Dear authors,
> >
> > Many thanks for your prompt replies. Most of my concerns were addressed. In particular, I find the new results including the attention mechanism in computational analysis and the new results on social networks interesting and important to add to the updated main text.
> >
> > One discussion I would like to put forward is why IGNN/EIGNN/CGS could not directly use the jumping knowledge connection, as one naive implementation is to stack several implicit layers and fuse hidden representations of them.

---

> > > ### Author Response · Authors · 2022-08-08
> > > **Further discussion on jumping knowledge connection**
> > >
> > > Really appreciate your reply and suggestions! Happy to see that most of your concerns were addressed. We will include those new results in the later version of our paper.
> > >
> > > We see your point about IGNN/EIGNN/CGS with jumping knowledge connection. We would like to point out that stacking several implicit layers cannot create extra representational power and capture multiscale information. The NeurIPS 2020 tutorial: Deep Implicit Layers by Zico Kolter (http://implicit-layers-tutorial.org/deep_equilibrium_models/, see the section "One (implicit) layer is all you need") demonstrates that a single implicit layer is equivalent to modelling any number of stacked implicit layers. The formal proof and theorem can also be found in DEQ paper [1] (see Theorem 2).
> > >
> > > Therefore, if we stack several implicit layers with the same single scale and fuse equilibriums of them, the model still cannot capture multiscale information since these stacked implicit layers would have the same neighborhood range. In contrast, in JKNet, the hidden representations at different explicit layers actually have different neighborhood ranges. We will add the above discussion to the future version of our paper.
> > >
> > > Hope our response clarifies your concern. If you are satisfied and think our submission with these new results and discussions is worth publishing, we hope you may consider slightly raising the score. We are willing to discuss/clarify further if you still have any concerns.
> > >
> > >
> > > [1] Shaojie Bai, J. Zico Kolter, and Vladlen Koltun. Deep equilibrium models. NeurIPS 2019. https://arxiv.org/pdf/1909.01377.pdf

---

> > > > ### Comment · Reviewer_Ctvi · 2022-08-08
> > > > **Thank you for the clarification**
> > > >
> > > > Thank you for this further explanation, which makes sense. I am happy to increase my score as Accept. Hopefully, the authors can revise their work accordingly based on all the comments of the reviewers.

---

> > > > > ### Author Response · Authors · 2022-08-09
> > > > > **Thank you for your reply**
> > > > >
> > > > > Thank you so much for your reply and your recognition! We are happy to see that your last concern has been addressed. We will definitely revise our work for the future/camera-ready version according to all the comments.
> > > > >
> > > > > Again, we really appreicate your help and suggestions during this discussion period. Your suggestions made our submission become a better one.

---

### Official Review · Reviewer_apvj · 2022-07-11

**Rating:** 6
**Confidence:** 4
**Soundness:** 3 good
**Presentation:** 3 good
**Contribution:** 3 good

**Summary:**

In this paper, the authors analyze the "infinite effect" in implicit GNNs, and give the theoretical bound of the "effective range". Additionally, the authors extend the implicit GNNs with multi-scale ability plus the "effective range" analysis. The whole paper is complete, and the experiment results are promising.

**Questions:**

1. For the first question, please refer to the first bullet point in the Limitations section.

2. In experiments, when a single scale is executed in MGNNI, is the attention mechanism in Eq. (7) - (10) still applied?

**Limitations:**

1. The definition of multi-scale is not clear and needs to be specified. In the current version, from lines 48 to 57, the authors listed four related works and illustrated two of them. Just describing them could not lead to a clear understanding of multi-scale, and more especially, why [8] is a multi-scale method? A more straightforward question here can be like: does stacking GNN layers qualify as a multi-scale method? Or the scale of each node feature aggregation needs to be personalized could qualify as a multi-scale method?

2. Authors may want to be more careful about the statements from lines 38 to 47. A suggestion here is to highlight and well-define the term "effective". Because of the Theorem 1 and its proof, the effect still exists even though it is bounded by a small value, and the threshold $\theta$ is hand-crafted.

3. I am wondering about the role of the attention mechanism in outstanding experimental performance. The ablation studies could better consider removing the attention part.

**Strengths And Weaknesses:**

Strengths
1. The flow of the paper is clear and well-written.
2. This paper gives a solid analysis of the implicit GNNs for their effective range. For dealing with the multi-scale problem for implicit GNNs, this paper proposes the MGNNI model with theoretical analysis.
3. The experiments are extensive.

Weaknesses
1. Some parts of paper can be reconsidered like the definition, derivation, and the experiments. For details and improvement suggestions, please refer to the following "Limitations" section.

---

> ### Author Response · Authors · 2022-08-02
> **Response to Reviewer apvj**
>
> We thank reviewer apvj for the valuable feedback and suggestions. We post our response below.
>
> **Q1.** "The definition of multi-scale is not clear and needs to be specified..."
>
> In our opinion, the key point to be considered as a multi-scale method is that the model should be able to adaptively adjust the neighborhood size or learn how to adaptively mix information from information at various distances. Note that for this definition/concept in our paper, we mainly follow the description in JKNet [30], N-GCN [2], and MixHop [1]. In addition, intuitively, multiscale can be somewhat considered as similar to different kernel sizes of Convolutional Neural Network.
>
> GCNII [8] is not a multi-scale method and stacking GNN layers doesn't qualify as a multi-scale method. The reason is that, given a stacked k-layer GNN model, for each node, information from 1-hop neighborhood are only aggregated indiscriminately at each layer and also the neighborhood size is fixed (i.e., k). Thus, stacking GNN layers cannot adaptively adjust the neighborhood size for each node. Therefore, GCNNI [8] and stacking GNN should be not considered as multi-scale methods.
>
> We will remove [8] from the corresponding sentence and revise the corresponding part of the paper in our final version.
>
> **Q2.** "Authors may want to be more careful about the statements from lines 38 to 47. A suggestion here is to highlight and well-define the term "effective"."
>
> Thanks for the suggestion. We would like to revise the term "effective range" to "$\theta$-effective range". We define $\theta$-effective range as follows:
>
>
> **Definition 1.** For any given error parameter $\theta > 0$, the $\theta$-effective range $h$ is the maximum integer such that exists some pairs of nodes $p$ and $q$ that are $h$-hop apart, when we perturb node features $X_{:,p}$ of node $p$ by some $\Delta X_{:, p}$, the L2 norm of the change in node $q$’s equilibrium $\left\lVert \Delta Z^{*}_{:,q}\right\rVert > \theta $.
>
> By the above definition, we can get that, given any $h' > h$, for **all pairs** of node $p$ and $q$ that are $h'$-hop apart, $\left\lVert \Delta Z^{*}_{:,q}\right\rVert \leq \theta $. Combining this with Equation (3) in Theorem 1, we can obtain the revised corollary 1 as follows:
>
> **Corollary 1.** With Equation (2) for propagation, given any error constant $\theta > 0$, the $\theta$-effective range $h$ is upper bounded as follows:
> $h < \frac{\ln (\theta(1-\gamma))}{\ln \gamma}$.
>
> The core idea of the proof is similar to that in the original proof. We will provide the revised definition and the complete proof in the final version of our paper. Corollary 3 will be revised similarly as well.
>
> We look forward to your comments on this revised definition of the term "effective" and would like to know if you think it is better than the previous one.
>
>
> **Q3.** "The ablation studies could better consider removing the attention part."
>
> Thanks for the suggestions. For an additional ablation study, we removed the attention mechanism and instead use average pooling for fusing information from multiple scales. The experimental results are follows (w/ att stands for "with attention", w/o att: "without attention"):
>
>
> | Scales            | PPI   | Chameleon | Texas |
> | ----------------- | ----- | --------- | ----- |
> | (w/ att) {1,2}    | 98.67 | 63.93     | 83.24 |
> | (w/ att) {1,2,3}  | 98.74 | 63.75     | 84.86 |
> | (w/o att) {1,2}   | 98.27 | 60.43     | 82.67 |
> | (w/o att) {1,2,3} | 98.36 | 59.17     | 83.21 |
>
> We can see that, if we replace the attention mechanism with average pooling, the performance would drop. It verifies the effectiveness of our attention mechanism, which is also demonstrated in Figure 3 and its corresponding explanations in our submission.
>
> **Q4.** "In experiments, when a single scale is executed in MGNNI, is the attention mechanism in Eq. (7) - (10) still applied?"
>
> No. If a single scale is executed, the attention mechanism should not be applied. The reason is that, in this case, there is no need to apply different attention weights on different scales. And if we still apply the attention mechanism nevertheless for a single scale, the attention weight would be always 1.0.

---

> ### Author Response · Authors · 2022-08-07
> **Look forward to your reply**
>
> Dear Reviewer apvj,
>
> First, we appreciate your insightful feedback and suggestions again. We would like to kindly remind you that, following your suggestions, we have revised/reconsidered some definitions and statements. We have also provided more ablation studies as suggested.
>
> Hope you are satisfied with our response. And we are looking forward to your reply. If you still have any concerns, we are also willing to discuss/clarify further.

---

> > ### Comment · Reviewer_apvj · 2022-08-09
> > **Reply to Authors of Paper8681**
> >
> > Dear Authors,
> >
> > Thanks for your reply to answer my questions, your paper will go into the Reviewer- Metareviewer Discussion period (08/09 to 08/19). Thanks again!

---

> > > ### Author Response · Authors · 2022-08-09
> > > **Thank you for your reply**
> > >
> > > Dear Reviewer apvj,
> > >
> > > Thank you so much for your reply. We really appreciate your insightful comments and suggestions again. We will revise our paper for the future/camera-ready version according to all the comments.

---

### Official Review · Reviewer_wdFc · 2022-07-12

**Rating:** 7
**Confidence:** 4
**Soundness:** 4 excellent
**Presentation:** 4 excellent
**Contribution:** 3 good

**Summary:**

The paper points out a main draw back of existing implicit graph neural network — its limited effective range. Although implicit graph neural network has implicitly infinite depth and infinite large reception field, the influence between two nodes decays exponentially with distance along the graph, so it still cannot capture long range information well. The paper propose to alleviate the problem by multi-scale propagation, which utilize adjacency matrix with different powers to propagate info, and finally fuse propagation results of different scales by attention mechanism.

**Questions:**

## About theorem 3:
1. eq. (11) has nothing to do with function $f(X, \mathcal{G})$ in eq. (4), which is strange. If I am correct, only when $f(X, \mathcal{G})=X$, theorem 3 holds.
2. If we replace $S$ in Theorem 1 by $S^m$, we can get following upper bound:
$$\lVert\Delta Z^*_{:,q}\rVert \le \frac{\gamma^h}{1-\gamma}\lVert g^h(F)\Delta X_{:,P} S^{hm}_{p,q}\rVert$$
- It seems to be a tighter bound than theorem 3 (I am not sure), because $\gamma, g(F)$ and $S_{p,q}$ are less than 1, above formula has higher power on them, and thus the upper bound is tighter.
- If above upper bound is tighter, then Corollary 2 will become same as Corollary 1, and thus the effective range remains unchanged even with multi-scale propagation. If am wrong, please correct me.

## About motivation:

Do we really need so called "long range information" in real world applications?
- In my opinion, locality is is not a drawback of Graph Neural Networks, or more generally, it is not a drawback of convolutional neural networks, including both CNN in image domain and GCN in graph domain. Instead, it is the key why CNN outperforms MLP and why GCN works. Nearby neighbourhood always has much more importance than nodes/pixels far away, because they are the nodes that closely related to the centre nodes/pixels.
- Even if someone take it as a drawback, their methods still assign more importance to nearby neighbourhood. Nobody tries to give more importance to nodes far away than nearby ones, because that is not reasonable. Take the proposed method as an example, the influence between two nodes still decays exponentially with distance.
- Experimental results also favours my opinion. In real word dataset, the proposed method performs similar as EIGNN, which does not consider multi-scale propagation. Only in handcrafted dataset designed for long range information, the proposed method outperforms EIGNN by a large margin.

This part about motivation is not a question, but a discussion. It is OK for you not to response.

**Ethics Review Area:**

["I don’t know"]

**Limitations:**

No special negative societal impact.

**Strengths And Weaknesses:**

Strength:
1. the paper is well motivated, well written and easy to follow.
2. The drawback of IGNN is demonstrated by both theoretical analysis and experimental proofs.
3. The effectiveness of proposed method is also demonstrated by both theoretical analysis and experimental proofs.

Weakness:
1. The modification does not thoroughly overcome the drawback, but just alleviate it instead. Although effective range becomes $m$ times larger, the influence between two nodes still decays exponentially with distance along the graph in the proposed method.
2. The proposed methods outperforms EIGNN by a large margin only in synthesized data, and behaves similarly as EIGNN in real world dataset.
3. I doubt the correctness and meaningfulness of some theorem, and expect explanations from authors. See my questions below.

---

> ### Author Response · Authors · 2022-08-02
> **Response to Reviewer wdFc**
>
> We thank reviewer wdFc for the valuable feedback and suggestions. We post our response below.
>
> **Q1.** "About Theorem 3: eq. (11) has nothing to do with function $f(X,G)$ in eq. (4), which is strange. If I am correct, only when $f(X,G) = X$, theorem 3 holds."
>
> Yes. In theorem 3, for simplicity and fair comparison with theorem 1, we let $f(X,G) = X$. We forgot to mention this in our submission and we will add it in our final version of the paper.
> The reason for letting $f(X,G) = X$ is to ensure all the conditions in theorem 1 and 3 are the same except for the multiscale propagation.
> Note that if needed, we can also use $f(X,G)$ as a more general form in both theorems and replace $X$ with $f(X,G)$ in Eq (2) for theorem 1.
>
> In our experiments, $f(X,G)$ is usually a MLP which has no effect on the effective range. Thus, if we keep $f(X,G)$ for both theorems and assume $f(X,G)$ itself has no effect on the effective range, it will lead to a similar conclusion that multiscale propagation provides a larger effective range.
>
> And take one more step further, if $f(X,G)$ itself can expand the effective range, the whole model will possibly have an even larger effective range than using $f(X,G)=X$. A new model/analysis regarding this might be interesting for future work.
>
> **Q2.** "If we replace $S$ in Theorem 1 by $S^m$, we can get the following upper bound: ..."
>
> We see your point. But we cannot directly replace $S$ with $S^m$ in the upper bound of theorem 1. The reason is that the upper bounds in theorem 1 and 3 are derived by using different propagations (i.e., Eq (2) for theorem 1 and Eq (4) with $f(X,G)=X$ for theorem 3). If we want to replace $S$ by $S^m$, we can only do it in the propagation formula first (i.e., Eq (2)). Then, it becomes exactly Eq (4) with $f(X,G)=X$. After that, following the process of the proof of theorem 3, we should still get the upper bound: $\left\lVert \Delta Z_{:, q}^{*}\right\rVert \leq \frac{\gamma^{\frac{h}{m}}}{1-\gamma}\left\lVert g^{\frac{h}{m}}(F) \Delta X_{:, p} S_{p, q}^{h}\right\rVert$.
>
> Hope this clarifies your concern. We are willing to clarify/discuss more if you still have concerns regarding this.
>
>
> **Q3.** "The proposed methods outperforms EIGNN by a large margin only in synthesized data, and behaves similarly as EIGNN in real world dataset."
>
> We agree that MGNNI outperforms EIGNN more significantly on synthetic datasets compared with on real-world datasets. But we also want to point out that MGNNI increases 8% absolute accuracy increase compared with EIGNN on Squirrel, and 2-3% on most datasets for graph classification.
>
>
>
> **Discussion about the motivation:**
>
> Actually, we are happy to see and get involved in the discussion about the need for long-range information, which we believe would be valuable for the community. We agree that nearby neighbourhood might be more important than nodes/pixels far away on some graphs. But we still believe that the *ability* of capturing long-range dependencies is important. However, it doesn't mean that we should always assign more importance to distant nodes. Instead, the model should have the *ability* to *adaptively* adjust where it can get valuable information (it can be short-range or long-range) and *adaptively* fuse/mix them for adapting to different graphs. The problem is that most traditional GNNs (e.g., GCN, GAT, SGC) *cannot* capture the long-range dependencies at all. So, our opinion is that the model should have the ability of capturing long-range dependencies and adaptively adjust the importance.
>
> Regarding the performance on real-world datasets in our experiments, please refer to our response for **Q3**.
>
> For other real-world applications, for instance, we think long-range information might be very useful in preventing further spread on disease spread chains. The incubation period of a disease can be long and during this period the disease might "propagates" far away through the chain to a very distant individual. If we can capture the long-range dependencies, we might early detect this potential infection instead of waiting until our own symptom or symptoms of nearby neighbors appeared. We think the synthetic datasets can somehow simulate this situation, although we don't have real-world datasets for this.

---

> > ### Comment · Reviewer_wdFc · 2022-08-08
> > **Thanks for your explanation.**
> >
> > I got it.
> >
> > If I take $S^m$ as the new adjacency matrix, then, in theorem 1, nodes $p$ and $q$ becomes $\frac{h}{m}$-hops away under the new adjacency matrix $S^m$.
> >
> > Let $h'=\frac{h}{m}$ and $S'=S^m$, and plug them into eq (3), then eq (3) becomes exactly same as eq (11). It seems that I found a shorter proof for theorem 3.
> >
> > Now, I am willing to endorse the technical soundness of the theorems, and give you an Accept.

---

> > > ### Author Response · Authors · 2022-08-08
> > > **Thank you for your reply**
> > >
> > > Dear Reviewer wdFc,
> > >
> > > Thank you so much for your reply and your recognition! We are happy to see that your concerns have been addressed. We agree that your proof for Theorem 3 is a shorter one.
> > >
> > > Again, we really appreciate your valuable comments and suggestions during this discussion period. We will revise and fix some small glitches as you suggested in the future version of our paper.

---

> ### Author Response · Authors · 2022-08-07
> **Look forward to your reply**
>
> Dear Reviewer wdFc,
>
> First, we appreciate your insightful feedback again. We would like to kindly remind you that we have posted our response to your concerns and questions about Theorem 3.
>
> Hope you are satisfied with our response. And we are looking forward to your reply. If you still have any concerns, we are also willing to discuss/clarify further.

---

### Official Review · Reviewer_Ap6e · 2022-07-12

**Rating:** 6
**Confidence:** 2
**Soundness:** 3 good
**Presentation:** 2 fair
**Contribution:** 2 fair

**Summary:**

In this paper, the authors propose implicit GNN with an effective range concept.

While previous implicit GNNs can be treated as neural networks with infinite layers sharing the same weights, MGNNI can exploit long-range dependencies capturing multi-scale information.

The $\gamma$ control can constrain the expressiveness of GNNs via effective range.

For multi-scale information, they exploit a set of scales $M$ and corresponding equilibriums $Z$.

The model is optimized in an iterative manner, not a closed-form solution because of the slow training speed over large graphs or a large number of node features.

MGNNI has comparable time complexity with IGNN and CGS because of the iterative method for convergence but it is more efficient than EIGNN which exploits eigendecomposition.

**Questions:**

-

**Limitations:**

The authors adequately addressed the limitations. While MGNNI can capture multi-scale information, there is a limitation from potential approximation error as other iteration-based methods such as IGNN and CGS.

**Strengths And Weaknesses:**

Implicit GNNs are getting more and more attention as one of the ways to complement the existing message-passing paradigm.

However, the researches for implicit GNNs are not plentiful that much compared to other explicit architectures.

The concept of effective range and the corresponding set of multiple scales with equilibriums might be akin to multi-hop aggregation of existing message passing GNNs such as ChebNet and SGC.

Using multiple propagation modules for multi-scale information may seem a bit naive, but I think it's well worth a try.

---

> ### Author Response · Authors · 2022-08-02
> **Response to Reviewer Ap6e**
>
> We appreciate reviewer Ap6e for the valuable feedback and recognition. We agree with you that capturing multi-scale information for implicit GNNs is worth trying. Considering there are only a few works on implicit GNN to date, We believe that our submission can be a valuable step on this topic. To point out again, our work provides the theoretical analysis on the effective range of implicit GNNs and proppose a new implicit GNN model which has an expanded effective range and can capture multi-scale information.

---

### Meta-Review · Area_Chair_Udin · 2022-08-24

**Recommendation:** Accept
**Confidence:** Less certain

**Metareview:**

The paper points out the limited effective range problem of existing implicit graph neural networks and proposes to use multi-scale propagation (i.e., use adjacency matrix with different powers) to alleviate the problem. All the reviewers vote towards an accept after the author rebuttal. The authors are encouraged to include content from the rebuttal into the camera ready version.

**Award:**

No

---

### Decision · Program_Chairs · 2022-09-14

Accept